# Health hazards related to using masks and/or personal protective equipment among physicians working in public hospitals in Dhaka: A cross-sectional study

Reaz Mahmud[1]*, K. M. Nazmul Islam Joy[2], Mohammad Aftab Rassel[3], Farhana Binte Monayem[4], Ponkaj Kanti Datta[5], Mohammad Sharif Hossain[6], Mohammad Mahfuzul Hoque[5], S. M. Habibur Rahman Habib[7], Nazmul Hoque Munna[8], Mohiuddin Ahmed[1], S. K. Jakaria Been Sayeed[3], Motlabur Rahman[5], Ahmed Hossain Chowdhury[1], Mohammad Zaid Hossain[5], Kazi Gias Uddin Ahmed[1], Md. Titu Miah[5], Md. Mujibur Rahman[9]

1 Department of Neurology, Dhaka Medical College, Dhaka, Bangladesh, 2 Shaheed Suhrawardy Medical College, Dhaka, Bangladesh, 3 Department of Neurology, National Institute of Neurosciences and Hospital, Dhaka, Bangladesh, 4 Medical officer, Sarkari karmachari Hospital, Dhaka, Bangladesh, 5 Department of Medicine, Dhaka Medical College, Dhaka, Bangladesh, 6 Research Investigator, Infectious Diseases Division (IDD), icddr,b, Dhaka, Bangladesh, 7 DNCC Dedicated Covid-19 Hospital Dhaka, Dhaka, Bangladesh, 8 Assistant professor, Department of Neurology, Mugda Medical College, Dhaka, Bangladesh, 9 Department of Medicine, Bangabandhu Sheikh Mujib Medical University, Dhaka. Bangladesh

* reazdmc22@yahoo.com, rmahmudneuro@dmc.gov.bd

**Data Availability Statement:** All relevant data are within the paper and its Supporting information files.

## Abstract

### Background

Wearing masks or personal protective equipment (PPE) has become an integral part of the occupational life of physicians due to the coronavirus disease 2019 (COVID-19) pandemic. Most physicians have been developing various health hazards related to the use of different protective gears. This study aimed to determine the burden and spectrum of various health hazards associated with using masks or PPE and their associated risk factors.

### Methods

This cross-sectional survey was conducted in Dhaka Medical College from March 01–May 30, 2021, among physicians from different public hospitals in Dhaka, Bangladesh. We analyzed the responses of 506 physicians who completed case record forms through Google forms or hard copies.

### Findings

The mean (SD) age of the respondents was 35.4 [7.7], and 69.4% were men. Approximately 40% were using full PPE, and 55% were using N-95 masks. A total of 489 (96.6%) patients experienced at least one health hazard. The reported severe health hazards were syncope, severe dyspnea, severe chest pain, and anaphylaxis. Headache, dizziness, mood irritation, chest pain, excessive sweating, panic attack, and permanent facial disfigurement were the

**Funding:** The research project received a grant of 2000 $ from Directorate General of Health Services, Government of People's Republic of Bangladesh. None of the author received any honorarium. The funders had no role in study design, data collection and analysis, decision to publish, or preparation of the manuscript.

**Competing interests:** The authors have declared that no competing interests exist.

minor health hazards reported. Extended periods of work in the COVID-19-unit, reuse of masks, diabetes, obesity, and mental stress were risk factors for dyspnea. The risk factors for headaches were female sex, diabetes, and previous primary headaches. Furthermore, female sex and reusing masks for an extended period (> 6 h) were risk factors for facial disfigurement. The risk factors for excessive sweating were female sex and additional evening office practice for an extended period.

## Conclusions

Healthcare workers experienced several occupational hazards after using masks and PPE. Therefore, an appropriate policy is required to reduce such risks.

## Introduction

The coronavirus disease 2019 (COVID-19), caused by severe acute respiratory syndrome coronavirus 2 (SARS CoV-2) [1], was declared a public health emergency of international concern by the Director-General of the WHO on January 30, 2020 [2]. As it was highly contagious, it rapidly became a pandemic [2,3]. Most countries experienced several waves of the infection with different SARS CoV-2 variants [4,5]. Therefore, the health systems of most countries have been exhausted due to the enormous burden of the disease.

As frontliners in the control of this pandemic, healthcare workers (HCWs) have been fatigued with the substantial task of diagnosing and treating the exponentially growing number of critical patients. According to a report by Amnesty International, approximately 7000 health workers died of COVID-19 by September 3, 2020 [6]. In January 2021, the WHO reported that 1.29 million HCWs were affected, which accounted for 8% of the cases [7]. The infection rate was 10% in the initial three months and it decreased to 2.5% by September 2020 [7]. Reported death to WHO COVID-19 surveillance in health care workers due to COVID -19, from January 2020–May 2021 was only 6643. Still, a population-based estimate revealed that 115 493 health and care workers could have died from COVID-19 during that period [8]. To decrease the infection rate among HCWs, the WHO released interim guidance [9], where this global health regulating body recommended using personal protective equipment (PPE) and N-95-like masks in settings where aerosol-generating procedures are frequent and only medical masks for other instances [9]. However, wearing these protective gears has physiological and psychological impacts [10]. In the previous epidemic of SARS, protective gear-induced health hazards led to the suboptimal use of the N-95 mask [11,12]. In the recent pandemic era, protective gear-related occupational hazards have not been well addressed and quantified in the scientific literature, despite the significant suffering of physicians and other HCWs reported by some small-scale studies conducted in different parts of the world [13–15]. To date, there have been no such studies in Bangladesh. The reported hazards include headache, skin breakdown, rash, impaired cognition, sweating, dry mouth, and dyspnea [13–15]. The prevalence of hazards ranged from 66–72% in different studies [13,14]. However, studies on risk factors are limited. Prolonged use (>4 h) of masks in those with previous primary headache was related to different hazards [13,14].

Physicians in Bangladesh need to wear masks or PPE during their COVID-19 duty period and in their regular evening office practice. Most doctors need to wear these devices frequently for extended periods due to the increasing workload, as doctors are scarce in Bangladesh. We should address every aspect of their suffering in order to acknowledge their roles. This study

aimed to determine the essence and spectrum of different health hazards related to the use of masks or PPE and their associated risk factors, which will make physicians as well as other HCWs aware of the health hazards. The identified risk factors will help to reduce the development of specific hazards. Furthermore, it will also increase the awareness in the concerned authorities of the burden of the problems and their attributed risks. Thus, this study will help in planning future action and preparing guidelines to reduce such occupational hazards.

## Methods

This study aimed to determine the burden and spectrum of various health hazards associated with using masks or PPE and their associated risk factors among physicians. We also assessed the severities of selected ailments and their impact on daily life.

### Study area and period

The study was conducted at Dhaka Medical College Hospital from March 01, 2021, to May 30, 2021. The respondents were physicians who had worked at least up to February 28, 2021, in the Dhaka Medical College Hospital, National Institute of Neurosciences and Hospital, Kurmitola General Hospital, Mugda Medical College and Hospital, and DNCC Dedicated COVID-19 Hospital. Among nine large COVID-19 dedicated hospitals in Dhaka city, we purposively selected these five hospitals (A map has been attached as Supporting information).

### Study design

This was a multi-centered cross-sectional survey. We sent a Google form (by email) or a hard copy of the case record form to the physicians (to be filled out by the respondents) according to the feasibility and availability of the responding physicians.

### Source population

Registered physicians (with qualifications of MBBS and higher) of all ranks who were in the duty roster of the above-mentioned hospitals, irrespective of the pattern of duty (COVID or non-COVID), whose phone number and e-mail number were available in the record of the hospital administration.

### Study population

The physicians who filled up their responses either on the Google form or in the hard copy of the case record form.

### Eligibility criteria

The physicians who consented to participate in the survey completely filled up the Google form or the hard copy of the case record form, and if all the information they provided was consistent, they were included in the survey. We excluded responders who provided incomplete or contradictory information. We also excluded physicians who were not using masks/respirators **due to contraindications**. We made the case record form in English and we used lot of medical terms. In Bangladesh we thought it would be difficult for the Nurses, pharmacist, midwiferies to understand every aspect of the case record form.

## Sample size determination

Sample size was determined using the formula [16] $\frac{z^2 pq}{d^2}$, where z = 1.96 (at 95% confidence level), p = 50% (as the prevalence of mask-induced hazards was not known in Bangladesh), and d = allowable error/precision = 5%. The estimated sample size was 384. We were not sure how many of the approached physicians would respond. So, we approached to 1122 physicians and 506 responses were ultimately included in the study.

## Sampling technique and procedure

We listed the names, e-mail addresses, and phone numbers of the physicians available in the roster of the above-mentioned hospitals and approached as many physicians as possible.

## Study variables

The variables included in the Google form/hardcopy of the case record form were.

**Demographic variable**: Age of the respondents in years and sex.

**Duty pattern**: Physician's position, pattern of duty (roster, morning, supervision, administrative), usual duty hours (8 hours, 12 hours, 8–12 hours), whether engaged in evening office practice.

**COVID-19-related questions**: COVID-19 positivity, symptoms in case of COVID-19 positivity (asymptomatic, mild, moderate, severe, critical).

**Protection-related questions**: Type of protection (full PPE, only mask), type of mask used (N-95, KN-95, surgical, homemade mask), frequency of mask wearing (daily, weekly, infrequently), whether masks are reused, duration of mask wearing (< 1 hour, 1–2 h, 3–4 h, 4–6 h, 6–8 h, >8 h), proper training on the use of PPE, condition of the doffing area (standard, average, below average).

**Outcome-related questions**: Experience of major events (syncopal attack, severe respiratory distress, severe chest pain, anaphylaxis), frequency of serious adverse effects (once, often, every time, infrequent), minor event after wearing a mask or PPE (headache, dizziness, irritation, exertional dyspnea, chest pain, excessive sweating, panic attack, disfigurement of the face, and others), frequencies of these experiences (infrequent, often, every time).

**Functional impact**: We assessed the functional effects of breathless and headache. Breathlessness was assessed according to the MRC score, including breathlessness on strenuous exercise or walking uphill, needing to slow down the pace of the walk, needing to stop walking after few minutes due to breathlessness, and breathlessness during routine walking. Headache impact was assessed using the HIT-6 score questionnaire comprising questions about the frequency of severe headaches, whether it limited daily activity, whether it made one wish to lie down, whether one tried to do work and got irritated, and whether it affected the ability to concentrate.

## Operational definition

Physicians were defined as those who had MBBS (or higher) qualifications and were registered with the Bangladesh Medical & Dental Council (BMDC). A mask was defined as any medical mask used for the personal protection of doctors against SARS-CoV-2, which included: 1. N-95 (3M) models like 8210, 1860; 2. 3M full face/half-face respirators; 3. layered surgical masks; and, 4. homemade masks. We defined PPE as a mask along with a coverall and face shield or goggles [10]. A COVID-19-dedicated hospital or unit referred to a hospital or a unit of hospitals designated by the government for the sole treatment of COVID-19-affected patients. We described the positions of physicians, their roles in clinical practice, and their duty patterns as

per job description defined by directorate general of health, Bangladesh [17]. We defined COVID-19 disease severity and presentation according to the WHO and Bangladesh guidelines on COVID-19 [18,19]. We described dyspnea as a feeling of shortness of breath either at rest or with a different grade of exertion and used an MRC score for grading it [20]. We defined headache as pain or any discomfort located in the head, excluding the face below the orbitomeatal line, including the nuchal ridge [21]. We described primary headache disorder as headache, or a headache disorder, not caused by or attributed to another disorder. It included tension-type, migraine, and cluster headaches [21]. We used the HIT-6 score questionnaire [22] to determine the functional impact of headaches. In this study, we defined syncope as a brief loss of consciousness attributed to cerebral hypoperfusion with a subsequent return to normal [23]. We described anaphylaxis as the constellation of several signs and symptoms caused by exposure to a provoking agent that occurs instantly [24]. We defined dizziness as unsteadiness other than vertigo and syncope. We defined severe adverse effects as symptoms that can endanger life or make an individual bedbound and enforce him or her to remove the mask or PPE. We described irritability as abnormal or excessive sensitivity or behavioral responses to environmental, situational, and emotional stimuli, and excessive sweating as unexpected abrupt sweating. Further, we defined a panic attack as a feeling of sudden, brief, and intense fear or apprehension without an apparent reason, and facial disfigurement as a persistent alteration in the face in the form of a change in color or scarring. Additionally, we considered severe respiratory distress, severe headache, anaphylaxis, severe chest pain, and syncope as serious adverse events. Minor adverse events included headache, dizziness, chest pain, irritation, sweating, panic attack, and facial disfigurement due to changes in the skin color. We gave the participants the option to note the adverse events they had developed in addition to these events.

COVID-19 units or hospitals in Dhaka City comprised four sections: an outpatient unit called a triage section, confirmed-COVID-19 wards, suspected-COVID-19 wards, and an intensive care unit (ICU). The patient first attended the triage section of the hospital. Medical officers and residents of different postgraduate subjects scrutinized the patients. They had an 8-hour roster duty. They used PPE with N-95/KN-95 masks during their total duty period, with some exceptions. Those who received patients in the ward or ICU used the same protection for the same duration. Those who worked in the suspected-COVID-19 sections sometimes took few safety precautions. Assistant professors or junior consultants were mainly involved in clinical rounds. They needed to take full precautions for 3–4 hours, and they also had evening office practice. Associate professors, senior consultants, and professors remained on call, and they rarely needed to wear full PPE.

### Data collection instrument

The data was collected either via the Google form sent via email or in the hard copy converted from the Google form (S1 File). The components of the questionnaire are described above in the variables section. We used a frequency scale for quantifying the outcomes. The questionnaire was developed by the authors as a Google form. Initial piloting for checking for internal consistency was done with 20 respondents, who were thereafter included in the research.

### Data collection procedure

We made a list of physicians with their phone numbers and e-mail addresses. Initially, we approached them by email. Those who did not respond were given a second reminder and were requested to respond within 15 days. Later, we made phone calls to the non-responding

physicians. We fixed appointments and approached those who gave verbal consent with a hard copy. Consenting physicians filled up the case record forms.

## Data quality control

The Google form was password-protected; therefore, only the principal investigator had access to the responses made by the respondents. We conducted a reliability analysis using Cronbach alpha of 0.7. We also conducted a validity analysis of the questionnaire using Pearson correlation (S 2). In the Google form, the respondents were unable to proceed to the next section without filling the mandatory fields. In case of those who filled up the case record form, trained data collectors were present while filling up the case record form, and they ensured that the mandatory fields had been completed. Every data sheet was reviewed by two researchers; in case of any differences, it was solved with discussion. No specific management was performed for missing data. In case of conflicting answers from the respondents, we excluded the response. There was no scope of double responses in the Google form, as one could respond only once from a specified email.

## Data processing and analysis

Data were analyzed using the Statistical Package for Social Sciences version 20 (IBM Corp. Armonk, NY, USA). We expressed the qualitative data as numbers and percentages, quantitative data with normal distribution as means (SD), and non-normal data as medians (IQR). We divided the respondents into groups who developed at least one hazard of any type, dyspnea of any severity, and headache of any severity. We compared them with the respondents who did not have any health hazards, dyspnea, or headache using the chi-square test. We used the unpaired t-test to test quantitative data with a normal distribution. We performed a binary logistic regression test to determine the risk factors for the development of headache, dyspnea, facial disfigurement, and excessive sweating. We described the odd ratio with 95% confidence intervals.

## Ethical consideration

The Institutional Ethical Committee of Dhaka Medical College approved the study (ERC-DMC/ECC/2021/55), and all participants provided written informed consents.

## Results

We contacted 1122 doctors; among them, 534 responded, giving a response rate of 47.6%. We included 506 respondents for analysis after scrutiny (Fig 1).

We excluded those who did not complete the consent form and had given incomplete or contradictory responses. A total of 489 (96.6%) participants had at least one health hazard (Table 1). The mean (SD) age of the respondents was 35.4 (7.7), and 69.4% were men. A total of 327 (64%) worked in COVID-19-dedicated units of different public hospitals in Dhaka and were either medical officers or residents (50%). Two-thirds of the respondents were on the roster, and their working hours were 8 hours. Approximately 60% had additional evening office practice of a duration of 2–4 hours. Approximately 90% had to use masks daily (55%, N-95 mask) for an average duration of 4–8 hours (66%), and 40% used full PPE. The most common comorbid diseases were asthma, hypertension, diabetes, and obesity. Approximately 30% had previous primary headaches, and 17% were very stressed due to COVID-19 (Table 1).

Those who had experienced at least one health hazard were relatively younger (mean [SD] age-35.2 [7.2] years; p = 0.001). Those who worked in COVID-19-dedicated hospitals/units

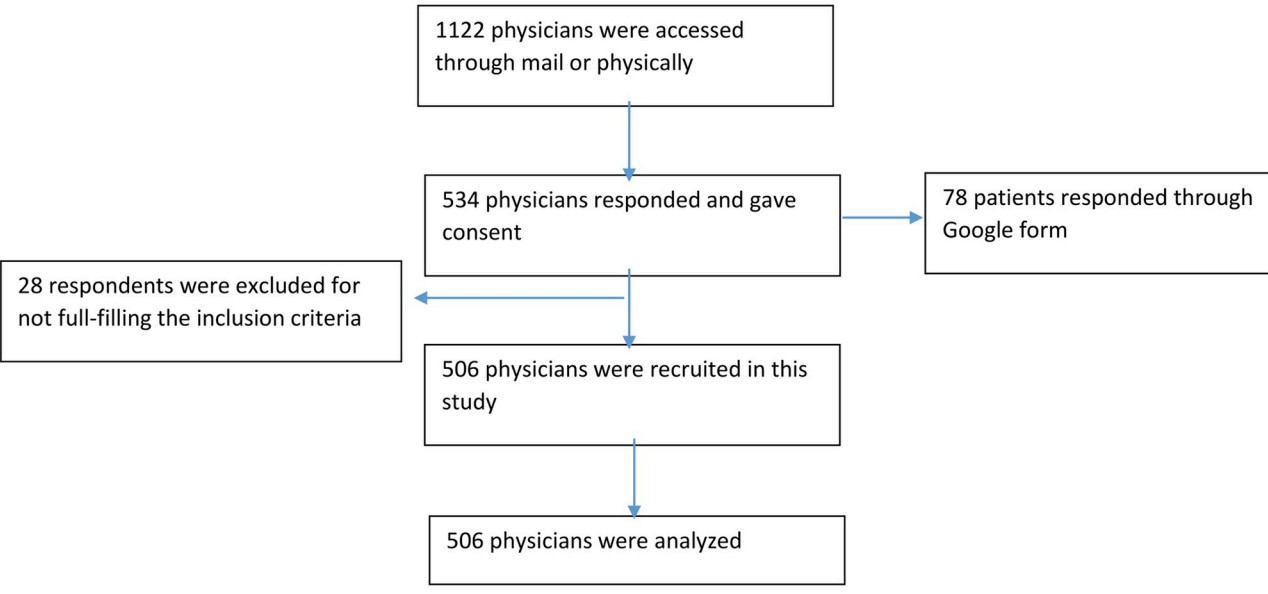

**Fig 1. Patient selection for the cross-sectional survey.**

(p-value 0.04) and reusing mask (p-value 0.04) were more likely to have experience at least one hazard. Furthermore, those who used only masks developed less hazards than those who used full PPE p-value 0.04) (Table 1).

Participants who had experienced respiratory distress of any severity were less than 45 years old (171 [39.1%]) (P-value-0.04). Moreover, working in the COVID-19-dedicated unit, reusing masks, and having bronchial asthma, diabetes, and obesity increased the risk of respiratory distress. Those whose duty hours were <8 hours were (p-value 0.003) less likely to have respiratory distress than those whose duty hours were >8 hours (Table 1).

More women experienced headaches ((p value <0.001). The use of full PPE, bronchial asthma, diabetes, and previous primary headache increased the risk of headache (Table 1).

Overall, 23 (5%) patients developed syncope; 96 (19%), severe respiratory distress; 32 (6%), severe chest pain; and, 6 (1%), anaphylaxis. Syncope (48%) and anaphylaxis (50%) occurred once in participants (48%). Severe respiratory distress (44%) and chest pain (72%) occurred infrequently (44%) (Fig 2a).

Overall, 377 (75%) patients developed headache; 143 (28%), dizziness; 232 (46%), irritation; 12 (2%), chest pain; 153 (30%), excessive sweating; and, 4%, panic attacks [Fig 2b]. In addition, 133 (26.3%) patients developed permanent facial disfigurement. Headache, excessive sweating (51%), and dizziness (36%) frequently occurred. In contrast, irritation (40%) and exertional dyspnea (46%) occurred infrequently (Fig 2b).

Most respondents reported respiratory distress on walking uphill (122 [52%]). Most respondents (148 [39%]) experienced severe headaches for some time. Approximately 86 (23%) respondents experienced severe headaches very often or always. The health hazards reported by participants included tiredness (104 [20.5%]), loss of concentration (142 [28%]), desire to lie down frequently (87 [17.2%]), and limitation in daily activity (142 [28.1%]) (Table 2).

Risk factors were analyzed using binary logistic regression. Initially, univariate logistic regression was performed (S2 Table), after which multivariate analysis with the forward conditional method was used. Working in COVID-19-dedicated units (OR, 95% CI, p = 1.3, [1.3–

**Table 1. Demography and distribution of different risk factors among the total population and the different subgroups.**

| Characteristics | Total n-506 | Presence of at least 1 hazard, N = 489 | p-value | Any type of dyspnea N = 207 | p-value | Headache n = 377 | p-value |
|---|---|---|---|---|---|---|---|
| Age mean (SD) | 35.4 (7.7) | 35.2 (7.4) | 0.01 | 35.9 (7.7) | 0.25 | 35.6 (7.3) | 0.3 |
| Age group | | | 0.05 | | 0.04 | | 0.3 |
| ≤ 45 years | 437 (86.4) | 425 (97.3) | | 171 (39.1) | | 329 (75.3) | |
| >45 years | 69 (13.6) | 64 (92.8) | | 36 (52.2) | | 48 (69.6) | |
| Gender | | | | | | | |
| Male, n (%) | 351 (69.4) | 339 (96.6) | 0.9 | 148 (42.2) | 0.38 | 245 (70.1) | <0.001 |
| Worked in COVID-19- dedicated hospital/unit[a], n (%) | 327 (64.6) | 320 (97.9) | 0.04, | 154 (47.1) | <0.001, | 260 (79.5) | 0.001 |
| Role in clinical practice | | | 0.001 | | 0.007 | | 0.3 |
| Supervision | 35 (6.9) | 30 (85.7) | | 15 (42.9) | | 23 (65.7) | |
| Clinical rounds | 309 (61.1) | 303 (98.1) | | 142 (46) | | 236 (76.4) | |
| Receiving and follow-up of the patient | 162 (32) | 256 (96.3) | | 50 (30.9) | | 118 (72.8) | |
| Duty pattern | | | 0.04 | | 0.9 | | 0.07 |
| Roster | 377 (74.5) | 367 (97.3) | | 155 (41) | | 284 (75.3) | |
| Morning | 73 (14.4) | 71 (97.3) | | 30 (41.1) | | 58 (79.5) | |
| On call | 56 (11.1) | 51 (81.1) | | 22 (37.3) | | 35(62.5) | |
| Duty hours | | | 0.8 | | 0.003 | | 0.6 |
| 8 hours | 375 (74.1) | 362 (96.5) | | 168 (44.8) | | 278 (73.9) | |
| 8–12 hours | 131 (25.9) | 127 (96.9) | | 39 (28.8) | | 100 (76.3) | |
| Evening office practice[i] (yes) | 320 (63.2) | 311 (97.2) | 0.38, | 126 (39.4) | 0.4 | 243 (75.6) | 0.33, |
| Duration of evening practice | | | 0.34 | | 0.03 | | 0.43 |
| 2 hours | 115 (22.7) | 109 (94.6) | | 45 (40.2) | | 86 (74.1) | |
| 3 hours | 43 (8.5) | 42 (97.6) | | 26 (56.5) | | 29 (67.4) | |
| 4 hours | 107 (21.1) | 105 (98.1) | | 32 (29.9) | | 83 (77.6) | |
| >4 hours | 55 (10.9) | 55 (100) | | 23 (41.8) | | 45 (81.8) | |
| COVID-19 status[c] | | | 0.37 | | 0.03 | | 0.3 |
| Positive | 187 (37) | 184 (98.4) | | 90 (48.1) | | 146 (78.1) | |
| Suspected | 35 (6.9) | 34 (97.1) | | 14 (40) | | 26 (74.3) | |
| Multiple infection | 4 (0.8) | 4 (100) | | 0(0) | | 4 (100) | |
| Protection used | | | 0.04 | | 0.01 | | 0.03 |

*(Continued)*

**Table 1.** (Continued)

| Characteristics | Total n-506 | Presence of at least 1 hazard, N = 489 | p-value | Any type of dyspnea N = 207 | p-value | Headache n = 377 | p-value |
|---|---|---|---|---|---|---|---|
| **Full PPE** [e] | 213 (42.1) | 210 (98.6) | | 101 (47.4) | | 169 (79.3) | |
| **Only mask** | 293 (57.9) | 279 (95.2) | | 106 (36.3) | | 268 (71) | |
| **Mask type** | | | 0.8 | | 0.025 | | 0.27 |
| N-95 or equivalents [f] | 280 (55.3) | 269 (96.1) | | 117 (41.8) | | 213(76) | |
| KN-95 | 55 (10.8) | 54(11) | | 31 (56.4) | | 43(78.2) | |
| Surgical mask | 161 (31.8) | 156(31.9) | | 57 (35.4) | | 113(70.2) | |
| Gas respirator [g] | 10 (2) | 10 (100) | | 2 (20) | | 8(80) | |
| **Reuse of the mask (yes)** | 272 (53.8) | 267 (98.2) | 0.04, | 132 (48.5) | <0.001 | 205 (75.4) | 0.6 |
| **Duration of mask wearing** | | | 0.7 | | 0.001 | | 0.6 |
| <1 hour | 10 (2) | 10 (100) | | 1 (10) | | 7 (70) | |
| 1–2 hours | 7 (1.4) | 7 (100) | | 2 (28.6) | | 5 (71.4) | |
| 3–4 hours | 20 (4) | 19 (95) | | 16 (80) | | 15 (75) | |
| 4–6 hours | 152 (30) | 148 (97.4) | | 53 (34.9) | | 116 (76.3) | |
| 6–8 hours | 183 (36.2) | 174 (95.1) | | 79 (43.2) | | 142 (77.6) | |
| >8 hour | 134 (26.5) | 131 (97.8) | | 56 (41.8) | | 92 (68.7) | |
| **Training on infection control measures, and donning and doffing (yes)** | 258 (51.1) | 255 (98.8) | 0.005 | 121 (46.9) | 0.005, | 201 (77.9) | 0.07 |
| **Comorbidity** | | | | | | | |
| **Asthma** | 72 (14) | 72 (100) | 0.06, | 42 (57.5) | 0.001 | 61 (85.9) | 0.02, |
| **Diabetes** | 40 (7.9) | 40 (100) | 0.2, | 23 (57.5) | 0.02, | 35 (87.5) | 0.04, |
| **Hypertension** | 65 (12.8) | 64 (98.5) | 0.7, | 24 (36.9) | 0.5 | 54 (83.1) | 0.08 |
| **Obesity** | 60 (11.5) | 56 (93.3) | 0.13, | 37 (61.7) | <001 | 38 (63.3) | 0.03 |
| **Previous primary headache** [h] | 156 (30.8) | 152 (97.4) | 0.5 | 65 (41.7) | 0.8 | 141 (90.4) | 0.001 |
| **Mental stress** [j] | | | 0.23 | | 0.001 | | 0.04 |
| **Highly stressed** | 90 (17.8) | 84 (93.3) | | 38 (42.2) | | 75 (83.3) | |
| **Worried but could cope up** | 280 (55.3) | 273 (97.5) | | 131 (46.6) | | 207 (73.9) | |
| **Slightly worried** | 119 (23.5) | 115 (96.6) | | 35 (29.4) | | 86 (72.3) | |

(*Continued*)

**Table 1.** (Continued)

| Characteristics | Total n- 506 | Presence of at least 1 hazard, N = 489 | p-value | Any type of dyspnea N = 207 | p-value | Headache n = 377 | p-value |
|---|---|---|---|---|---|---|---|
| **Not worried** | 17 (3.4) | 17 (100) | | 3 (17.6) | | 9 (52.9) | |

[a]-A COVID-19-dedicated hospital or unit refers to hospital or a unit of a hospital designated by the government for the sole treatment of COVID-19-affected patients.

[b]-Role in clinical practice and duty pattern were described as per job description defined by directorate general of health, Bangladesh.

[c]-As described by the case definition of WHO.

[d]-As described by WHO and Bangladesh guidelines.

[e]-a mask along with other protective equipment for protection of the other parts of the body, which includes a coverall and/or face shield or goggles.

[f]-N-95 (3M) models like 8210, 1860.

[g]-3M full-face/half-face respirators.

[h]-Primary headache disorder was defined as headache, or a headache disorder, not caused by or attributed to another disorder, such as tension-type headache, migraine, and cluster headaches.

[i]-Evening clinical practice in own office after the regular duty in the public hospital.

[j]-Stress is the inability to cope with mental pressure and being overwhelmed with anxiety.

COVID-19: Corona virus disease 19.

PPE: Personal Protective Equipment.

3], 0.002), extended working hours (OR, 95% CI, p-value 0.7, [0.5–0.9], 0.001), reusing masks (OR, 95% CI, p-value 1.7, [1.7–2.5], 0.007), presence of diabetes (OR, 95% CI, p-value 2.1, [1–4.2], 0.04), and obesity (OR, 95% CI, p-value 2.9, [1.6–5.2], 0.001). increased the chance of dyspnea of any severity. Th The physicians with low personal stress levels developed less dyspnoea (OR, 95% CI, p-value 0.7, [0.5–0.9], 0.01) (Table 3).

For headache, female sex (for male sex OR, 95% CI, p-value 0.4, [0.24–0.67], 0.001), working in a COVID-19-dedicated unit (OR, 95% CI, p = 2.01, [1.3–3.1], 0.002), presence of diabetes (OR, 95% CI, p-value 2.9, [1.1–8.1], 0.03), and previous primary headache (OR, 95% CI, p-value 4.7, [2.6–8.5], <0.001), increased the susceptibility to headache of any severity (Table 3).

Female sex (for male sex OR, 95% CI, p-value 0.4, [0.24–0.67], 0.001), having evening practice (for no practice OR, 95% CI, p-value 0.55, [0.35–0.84], 0.006), using protection of high level (for low-level protection OR, 95% CI, p-value 0.55, [0.34–0.86], 0.005), increasing frequency of mask use (OR, 95% CI, p-value 1.8, [1.2–2.8], <0.001), increasing duration of mask wearing (OR, 95% CI, p = 1.3, [1.1–1.6], 0.001), increased the prevalence of facial disfigurement of any type (Table 3).

Lower age group (for higher age OR, 95% CI, p-value 0.9, [0.9–0.99], 0.04), female sex (for male, OR, 95% CI, p-value 0.4, [0.28–0.7], 0.001), evening practice (for no practice OR, 95% CI, p-value 0.21, [0.09–0.5], 0.004), longer practice duration (OR, 95% CI, p-value 1.5, [1.2–2.1], 0.001), COVID-19 positivity (OR, 95% CI, p-value 1.8, [1.3–2.5], 0.001), using high-level protection (for low-level protection OR, 95% CI, p-value 0.18, [0.11–0.3], 0.001), mask type (OR, 95% CI, p-value 1.3, [1.1–1.8], 0.03), and hypertension (OR, 95% CI, p-value 2.4, [1.2–4.6], 0.007) were associated with the increased prevalence of excessive sweating (Table 3).

## Discussion

This study demonstrated the occupational hazards related to the use of masks or PPE among physicians. Few participants developed severe respiratory distress, syncope, severe chest pain, and anaphylaxis, which is concerning. A significant number of participants developed

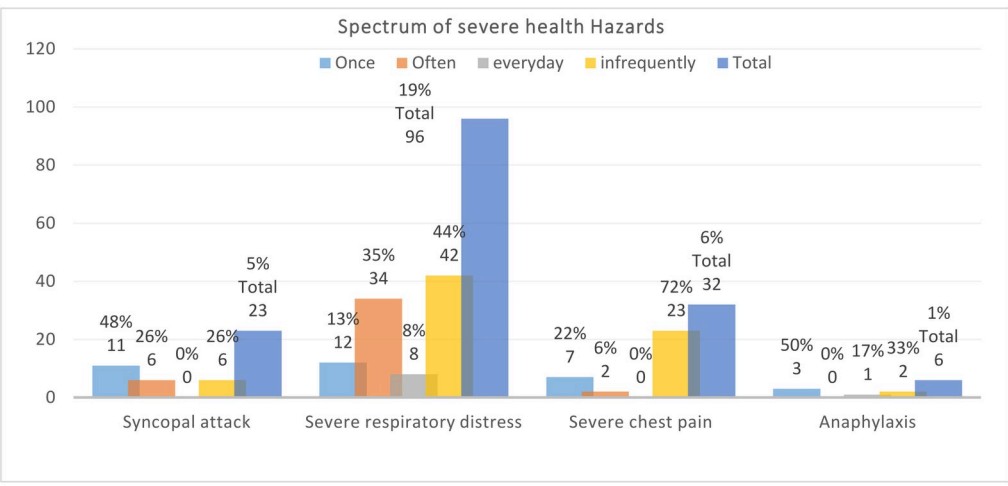

**Fig 2. The spectrum of overall health hazards associated with using masks or PPE.** a. spectrum of severe health hazards related to using masks and or PPE b. spectrum of minor hazards related to using of masks and or PPE.

headaches, dizziness, exertional dyspnea, mood irritation, excessive sweating, panic attacks, chest pain, and facial disfigurement. These "occupational hazards" have caused significant functional problems for a large number of physicians. The risk factors for the different health hazards were age group, female sex, diabetes, hypertension, mental stress, reuse of masks, wearing masks for a longer duration, using PPE, using KN-95 masks, and COVID-19 positivity.

This study was a novel approach to assess various health problems experienced by physicians due to the use of protective gear during the pandemic, in addition to being affected by the deadly virus. It would acknowledge their sacrifice during the pandemic. Although the

**Table 2. Severity (dyspnea and headache) and the functional impact of health hazards on the daily life of the physicians.**

| Trait | Number | Percentage |
|---|---|---|
| **Severity grade of dyspnea [a]** | | |
| Strenuous exercise | 27 | 24 |
| Walking uphill | 122 | 52 |
| Slowing down the pace | 28 | 12 |
| Stopping to walk | 11 | 5 |
| Breathless on routine exercise | 15 | 6 |
| **Experienced severe headache [b]** | | |
| Never | 58 | 15 |
| Rarely | 85 | 23 |
| Sometimes | 148 | 39 |
| Very often | 45 | 12 |
| Always | 41 | 11 |
| **Loss of concentration in the last 4 weeks** | | |
| Never | 261 | 51.6 |
| Rarely | 66 | 13 |
| Sometimes | 142 | 28 |
| Very often | 33 | 6.5 |
| Always | 4 | 0.8 |
| **Irritation in the last 4 weeks** | | |
| Never | 322 | 63.6 |
| Rarely | 64 | 12.6 |
| Sometimes | 88 | 17.4 |
| Very often | 29 | 5.7 |
| Always | 3 | 0.6 |
| **Tiredness in the last four weeks** | | |
| Never | 304 | 60 |
| Rarely | 67 | 13.2 |
| Sometimes | 104 | 20.5 |
| Very often | 29 | 5.7 |
| Always | 2 | 0.4 |
| **Desire to lie down frequently** | | |
| Never | 315 | 62.2 |
| Rarely | 55 | 10.9 |
| Sometimes | 87 | 17.2 |
| Very often | 34 | 6.7 |
| Always | 15 | 3 |
| **Limitation in daily activities** | | |
| Never | 176 | 34.8 |
| Rarely | 92 | 18.2 |
| Sometimes | 142 | 28.1 |
| Very often | 56 | 11.1 |
| Always | 40 | 7.9 |

[a]. According to the MRC scale.

[b]. According to the verbal rating scale.

**Table 3. Risk factors[a] of respiratory distress, headache, facial disfigurement, and excessive sweating.**

| Trait | Reference | B | SE | wald | P-value | OR | 95%CI |
|---|---|---|---|---|---|---|---|
| **Respiratory distress[b]** | | | | | | | |
| Working in COVID-19-dedicated units | Yes | 0.67 | 0.21 | 10.7 | 0.002 | 1.9 | 1.3–3 |
| Duty hours | <8 hour | -0.35 | 0.14 | 6.7 | 0.01 | 0.7 | 0.5–0.9 |
| Reuse of mask | Yes | 0.53 | 0.20 | 7.3 | 0.007 | 1.7 | 1.2–2.5 |
| Asthma | Yes | 0.74 | 0.28 | 7.01 | 0.008 | 2.1 | 1.2–3.6 |
| Diabetes | Yes | 0.73 | 0.36 | 4.2 | 0.04 | 2.1 | 1.0–4.2 |
| Obesity | Yes | 1.01 | 0.3 | 12.6 | 0.001 | 2.9 | 1.6–5.2 |
| Personal stress level | Low | -0.35 | 0.13 | 6.5 | 0.01 | 0.7 | 0.5–0.9 |
| Constant headache[c] | | -0.16 | 0.41 | 0.17 | 0.68 | 0.84 | |
| Sex | Male | -0.91 | 0.26 | 12.04 | 0.001 | 0.4 | 0.24–0.67 |
| Working in COVID-19-dedicated units | Yes | 0.69 | 0.22 | 9.8 | 0.002 | 2.01 | 1.3–3.1 |
| Diabetes | Yes | 1.08 | 0.51 | 4.5 | 0.03 | 2.9 | 1.1–8.1 |
| Stress level | Low | -0.35 | 0.15 | 5.7 | 0.01 | 0.7 | 0.52–0.9 |
| Previous primary headache | Yes | 1.5 | 0.3 | 26.5 | <0.001 | 4.7 | 2.6–8.5 |
| Constant | | 1.6 | 0.4 | 15.4 | <0.001 | 5.3 | |
| **Facial disfigurement[d]** | | | | | | | |
| Sex | Male | -0.91 | 0.26 | 12.04 | 0.001 | 0.4 | 0.24–0.67 |
| Evening practice | No | -0.59 | 0.21 | 7.6 | 0.006 | 0.55 | 0.35–0.84 |
| Protection used | Mask only | -0.6 | 0.21 | 7.7 | 0.005 | 0.55 | 0.34–0.84 |
| Frequency of mask use | Daily | 0.56 | 0.24 | 5.7 | 0.02 | 1.8 | 1.1–2.8 |
| Reuse of mask | Yes | 0.58 | 0.2 | 6.9 | 0.02 | 1.8 | 1.2–2.8 |
| Duration of mask wearing | >6 hours | 0.27 | 0.11 | 6.01 | 0.014 | 1.3 | 1.1–1.6 |
| Constant | | -2.6 | 0.8 | 11.2 | 0.001 | 0.07 | |
| **Excessive sweating[e]** | | | | | | | |
| Age | | -0.03 | 0.02 | 4.1 | 0.04 | 0.9 | 0.9–0.99 |
| Sex | Male | -0.8 | 0.2 | 11.7 | 0.001 | 0.4 | 0.28–0.7 |
| Duty hours | <8 hour | -0.6 | 0.2 | 9.4 | 0.002 | 0.5 | 0.4–0.8 |
| Evening office practice | No | -1.5 | 0.4 | 14.8 | 0.001 | 0.21 | 0.09–0.5 |
| Practice duration | >4 hours | 0.4 | 0.14 | 9.9 | 0.002 | 1.5 | 1.2–2.1 |
| COVID-19 positivity | Positive | 0.6 | 0.16 | 13.1 | 0.001 | 1.8 | 1.3–2.5 |
| Protection used | Mask only | -1.7 | 0.26 | 41.8 | 0.001 | 0.18 | 0.11–0.3 |
| Mask Type | Non-Filtering | 0.30 | 0.14 | 4.8 | 0.03 | 1.3 | 1.1–1.8 |
| Hypertension | Yes | 0.89 | 0.33 | 7.4 | 0.007 | 2.4 | 1.2–4.6 |
| constant | | 3.5 | 0.8 | 20.1 | 0.001 | 34.9 | |

[a]. Analyzed with multivariate binary logistic regression; forward conditional methods were used.

[b]. Omnibus test of model coefficients, 0.000; Nagelkerke R, 19.9; Hosmer and Lemeshow test, 0.48; sensitivity, 67.8; conditional forward method; model at step 7.

[c]. Omnibus test of model coefficients, 0.000; Nagelkerke R, 18.8; Hosmer and Lemeshow test, 0.63; sensitivity, 75.2%; conditional forward method; model at step 5.

[d]. Omnibus test of model coefficients, 0.000, Nagelkerke R, 13.8; Hosmer and Lemeshow test, 0.82; sensitivity, 75.1%; conditional forward method; model at step 5.

[e]. Omnibus test of model coefficients, 0.000; Nagelkerke R, 32.1; Hosmer and Lemeshow test, 0.16; sensitivity, 78.1%; conditional forward method; model at step 9.

COVID-19: Corona virus disease 19.

OR: Odd ratio.

SE: Standard error.

WHO recommends the use of filtering masks in aerosol-generating areas [10], many physicians are using filtering masks in general OPD patient consultations and indoor environments with the risk of COVID-19 because of the lack of capacity to conduct RT-PCR tests in all patients, as well as the negligence of people in using masks. Therefore, users of filtering masks

outnumbered the physicians working in COVID-19-dedicated units (65% vs. 70%). The number of physicians with at least some degree of stress in this study was high (approximately 70%), consistent with the results of previous studies on the psychological assessment of doctors (50%) [25]. Despite using filtering masks, a number of physicians had either confirmed or suspected COVID-19 infections (~44%) and moderate-to-severe disease. The prevalence was higher than that reported in other studies (5–21%) [26–29]. The increased prevalence in this study might be due to the lack of training, reuse of masks, and extended duty hours. Furthermore, a quarter of the physicians needed to wear masks for more than eight hours due to the shortage of doctors in many hospitals, although the WHO recommends using masks or PPE for 6 hours and reusing them only after reprocessing [10].

The physicians in this study experienced several health hazards similar to those in other studies [30,31]. Among the health hazards, dyspnea is the most concerning, which might be due to increasing resistance during inhalation, the addition of 50–100 ml dead space (the mask area), and re-breathing of a small volume of exhaled gas within that space [32]. Furthermore, wearing the mask can increase end-tidal $CO_2$ levels [12,33,34]. Filtering masks usually offer more resistance than surgical masks [32], and doctors using filtering masks reported more respiratory distress than others in this study. Some studies conducted in the pre-COVID-19 era found that filtering masks did not impose any respiratory distress when worn for 1 h [35]. However, in the COVID era, physicians need to wear filtering masks for extended periods. Patients with obstructive pulmonary disease with modified Medical Research Council dyspnea scale scores >3 or FEV1 < 30% predicted should be cautious while wearing masks or PPE [36]. Filtering masks increase airway resistance to 126% and 122% during inspiration and expiration, respectively, and reduce the gas exchange volume by 37% in each breath [37]. Reuse of the filtering mask can cause pore clogging. Thus, it increases breathing resistance [10]. Furthermore, moisture increases airway resistance by 3% [38]. Therefore, the duration of wearing filtering masks should be limited to 6 h. One should not reuse masks, and patients with obstructive pulmonary disease should be cautious. According to this study, individuals with obesity and uncontrolled diabetes should be carefully monitored.

Headache was the second most common health hazard reported in this study (74%). Its prevalence was similar to that reported in other studies [13 (71.4%, New York), 28 (81%, Singapore)]. Little difference in the prevalence of headache was found among the different geographic locations, such as the USA (71%) [14], Singapore [81%], and Morocco (62%) [39]. Headache might be due to compression, hypercapnia, or stress. Headaches arise from the sustained pressure on pericranial soft tissues resulting from donning objects with tight bands or straps around the head (e.g., hat, helmet, goggles worn during swimming or diving, or frontal lux devices) and have been previously reported in the literature [30,40–45]. Furthermore, alterations in cerebral hemodynamics due to build-up of $CO_2$ after wearing filtering masks may also cause headaches [46]. A study in the USA reported that 25% of the respondents experienced a headache after 3 h of wearing a mask [13]. High mental stress (70%) and previous primary headache (30%) may have increased the risk of headaches in this study. Therefore, physicians must address the underlying primary headache.

Anaphylaxis is one of the rarest, but most serious, health hazards reported in this study. This is probably due to hypersensitivity to the components and chemicals used in masks or PPE. Several studies have reported contact dermatitis related to the use of protective gears [14,47–49]. However, these 1% cases were probably the first to be reported.

Syncope (5%) and dizziness (28%) reported in this study may be due to prolonged standing and N-95-induced respiratory alkalosis [50]. Hypocarbia may cause cerebral vasoconstriction. Low intracranial pressure and cardiac arrhythmia [50] may be involved in the underlying pathophysiology of syncope and dizziness.

Permanent facial disfigurement (26%) is an exigent cosmetic problem reported in this study; one study in the USA found skin problems in 51% of the respondents. This may be due to contact dermatitis, scaling, acne, skin breakdown, and other skin conditions that can occur with frequent mask usage [14] or reuse of unhygienic masks. Furthermore, wearing masks for long durations also changes the local natural skin milieu and leads to local rises in temperature and humidity. It may cause an increase in skin pH, redness of the skin, fluid loss, and increased sebum production [51].

In this study, we found that health hazards caused a significant number of physicians to become irritated and limit their daily activities. A recent review article also described the neurological and psychiatric impacts of filtering masks [51].

This study included an adequate number of participants; the nature of work of the study population was relatively homogenous. All participants were experts and knowledgeable in their respective fields. Therefore, it was easier for them to understand the technical language used in the questionnaires. The responses are expected to be valid, representative, and reliable. The validity test also reflected this scenario.

In this study, the participants themselves provided the responses. Therefore, there may have been inappropriate responses due to a lack of understanding and communication. The reported experiences were real-time due to the nature of the study; however, the results might change with time. Therefore, a prospective study is warranted. Again, in this study respond modalities are not similar to all respondents.

We conducted the study mainly among physicians of different government institutes in Bangladesh, a tropical country with a hot and humid environment, where most physicians need to work without air-conditioning systems, proper resting facilities, or adequate rehydration opportunities. Working patterns and workloads might differ in various circumstances and geographical locations. We conducted the study on the physicians only; duty patterns, awareness, and training might differ among the other healthcare workers. Therefore, these results cannot be generalized for all instances. A multicenter, multinational study, inclusive of all sectors of the health care providers, is required to generalize these finding.

## Conclusions

Healthcare workers in the pandemic era are experiencing several occupational hazards, especially headache, different degrees of dyspnea, facial disfigurement, and chest pain, which also cause some functional disabilities. The important risk factors identified for different hazards were female sex and the presence of comorbidities, such as hypertension and diabetes. Wearing masks for longer durations and reusing them were also risk factors for some hazards.

## Supporting information

**S1 Fig. A map showing the location of the hospitals.**
(JPG)

**S1 Table. Validity test.**
(XLS)

**S2 Table. Univariate logistic regression analysis of the risk factors of respiratory distress, headache, facial disfigurement, and excessive sweating.**
(DOCX)

**S1 File. Google form.**
(PDF)

**S2 File. Editage edited manuscript.**
(DOCX)

**S3 File. Editage certificate.**
(PDF)

**S1 Dataset. Data set.**
(XLSX)

## Acknowledgments

The physicians who participate in the study.

## Author Contributions

**Conceptualization:** Reaz Mahmud, K. M. Nazmul Islam Joy, Mohammad Aftab Rassel, Farhana Binte Monayem, Ponkaj Kanti Datta, Mohammad Sharif Hossain, Mohammad Mahfuzul Hoque, S. M. Habibur Rahman Habib, Nazmul Hoque Munna, Mohiuddin Ahmed, S. K. Jakaria Been Sayeed, Motlabur Rahman, Ahmed Hossain Chowdhury, Mohammad Zaid Hossain, Kazi Gias Uddin Ahmed, Md. Titu Miah, Md. Mujibur Rahman.

**Data curation:** Reaz Mahmud, K. M. Nazmul Islam Joy, Mohammad Aftab Rassel, Farhana Binte Monayem, Ponkaj Kanti Datta, Mohammad Sharif Hossain, Mohammad Mahfuzul Hoque, S. M. Habibur Rahman Habib, Nazmul Hoque Munna, Mohiuddin Ahmed, S. K. Jakaria Been Sayeed, Motlabur Rahman, Ahmed Hossain Chowdhury, Mohammad Zaid Hossain, Kazi Gias Uddin Ahmed, Md. Titu Miah, Md. Mujibur Rahman.

**Formal analysis:** Reaz Mahmud, K. M. Nazmul Islam Joy, Ponkaj Kanti Datta, Mohammad Sharif Hossain, Md. Mujibur Rahman.

**Funding acquisition:** Reaz Mahmud, Kazi Gias Uddin Ahmed, Md. Titu Miah.

**Investigation:** Reaz Mahmud, K. M. Nazmul Islam Joy, Mohammad Aftab Rassel, Farhana Binte Monayem, Ponkaj Kanti Datta, Mohammad Sharif Hossain, Mohammad Mahfuzul Hoque, S. M. Habibur Rahman Habib, Mohiuddin Ahmed, S. K. Jakaria Been Sayeed, Motlabur Rahman, Ahmed Hossain Chowdhury, Mohammad Zaid Hossain, Kazi Gias Uddin Ahmed, Md. Titu Miah, Md. Mujibur Rahman.

**Methodology:** Reaz Mahmud, K. M. Nazmul Islam Joy, Mohammad Aftab Rassel, Farhana Binte Monayem, Ponkaj Kanti Datta, Mohammad Sharif Hossain, Mohammad Mahfuzul Hoque, S. M. Habibur Rahman Habib, Nazmul Hoque Munna, Mohiuddin Ahmed, S. K. Jakaria Been Sayeed, Motlabur Rahman, Ahmed Hossain Chowdhury, Mohammad Zaid Hossain, Kazi Gias Uddin Ahmed, Md. Titu Miah, Md. Mujibur Rahman.

**Project administration:** Reaz Mahmud, Mohammad Sharif Hossain, Mohammad Mahfuzul Hoque, Motlabur Rahman, Ahmed Hossain Chowdhury, Kazi Gias Uddin Ahmed, Md. Titu Miah, Md. Mujibur Rahman.

**Resources:** Reaz Mahmud.

**Software:** Reaz Mahmud, K. M. Nazmul Islam Joy.

**Supervision:** Reaz Mahmud, Md. Mujibur Rahman.

**Validation:** Reaz Mahmud, Farhana Binte Monayem, Md. Mujibur Rahman.

**Visualization:** Reaz Mahmud, Md. Mujibur Rahman.

**Writing – original draft:** Reaz Mahmud, Farhana Binte Monayem.

**Writing – review & editing:** Reaz Mahmud, K. M. Nazmul Islam Joy, Mohammad Aftab Rassel, Farhana Binte Monayem, Ponkaj Kanti Datta, Mohammad Sharif Hossain, Mohammad Mahfuzul Hoque, S. M. Habibur Rahman Habib, Nazmul Hoque Munna, Mohiuddin Ahmed, S. K. Jakaria Been Sayeed, Motlabur Rahman, Ahmed Hossain Chowdhury, Mohammad Zaid Hossain, Kazi Gias Uddin Ahmed, Md. Titu Miah, Md. Mujibur Rahman.

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
