## [Decision Letter · Decision Letter 0]

17 Mar 2022

PONE-D-21-25497Health hazards related to using masks and, or personal protective equipment among the physicians working in different public hospitals in Dhaka: A cross-sectional studyPLOS ONE

Dear Dr. Reaz Mahmud ,

Thank you for submitting your manuscript to PLOS ONE. After careful consideration, we feel that it has merit but does not fully meet PLOS ONE’s publication criteria as it currently stands. Therefore, we invite you to submit a revised version of the manuscript that addresses the points raised during the review process.

ACADEMIC EDITOR Comments:Why you used the heading *“Methods and Materials”? *Is there a material that used for this study? if not, please omit the “materials”.The methods will be better if structered in this way; Study Area and Period, Study Design, source population, study population, Eligibility criteria, Sample Size Determination, Sampling Technique and Procedures, Study Variables, Operational Definitions, Data Collection Instrument, Data Collection Procedures, Data Quality Control, and Data Processing and Analysis. Specially, this section should be critically addressed!The authors should clearly explain their Eligibility criteria as; inclusion and exclusion criteria.The authors must operationalize their outcome variables. How did they measure their outcome variables?Concerning to the sample size calculation for this study, the authors explained as …. So the estimated sample size was 384. We included a total of 506 respondents in this study” why you didn’t used the calculated sample size? This needs a justification.How did they get the study subjects? They have to clearly address their sampling technique? Since you conducted the data collection with online means, how did you keep the data quality for this study?Study instrument: have you developed, adapted or adopted the questionnaires? You have to clearly explain about the questionnaires you used because this is crucial. You have to report the reliability analysis finding for your questionnaires.Add the response rate.**Discussion:** discuss by using the scientific reasoning.**Conclusion:** avoid numbering. Make it a paragraph. Focus on your main findings, clinical implications, recommendations etc.

A rebuttal letter that responds to each point raised by the academic editor and reviewer(s). You should upload this letter as a separate file labeled 'Response to Reviewers'.A marked-up copy of your manuscript that highlights changes made to the original version. You should upload this as a separate file labeled 'Revised Manuscript with Track Changes'.An unmarked version of your revised paper without tracked changes. You should upload this as a separate file labeled 'Manuscript'.Please include the following items when submitting your revised manuscript:

We look forward to receiving your revised manuscript.

Kind regards,

Addisu Dabi Wake

Academic Editor

PLOS ONE

“Directorate General of Health services, Government of the People’s Republic of Bangladesh”

“The research project received a grant of 2000 $ from Directorate General of Health Services, Government of People's Republic of Bangladesh.

None of the author received any honorarium.

he funders had no role in study design, data collection and analysis, decision to publish, or preparation of the manuscript”

Reviewers' comments:

Reviewer's Responses to Questions

**Comments to the Author**

1. Is the manuscript technically sound, and do the data support the conclusions?

Reviewer #1: Yes

Reviewer #2: Partly

2. Has the statistical analysis been performed appropriately and rigorously? 

Reviewer #1: No

Reviewer #2: Yes

3. Have the authors made all data underlying the findings in their manuscript fully available?

Reviewer #1: No

Reviewer #2: Yes

4. Is the manuscript presented in an intelligible fashion and written in standard English?

Reviewer #1: Yes

Reviewer #2: No

5. Review Comments to the Author

Reviewer #1: Manuscript Title: Health hazards related to using masks and, or personal protective equipment among the physicians working in different public hospitals in Dhaka: A cross-sectional survey

1. Title: it is researchable, interesting and targeted. The manuscript had presented Health hazards related to using masks and, or personal protective equipment among the physicians working in public hospitals. It can be a valuable scientific literature if published after incorporation of the comments raised.

2. Abstract:

-

3. Introduction

-

4. Methods

Better to state about the brief background of your study area, from how many hospitals the physicians recruited.

Why you conduct only on physicians, what about other health care workers like Nurses, pharmacists, midwiferies, Anesthesia…?

If you conceptual framework it is better to show the relationship of predictor and outcome variables in the form of conceptual framework.

Better to clarify about your sample size determination, I think you used census method, so what is the advantage of using single population proportion formula?

What types of technique you used to keep/control your data quality? Since data quality is the corner stone for mixed research approach.

I haven’t seen about your tool/questionnaire or checklist you used to collect your data; therefore, please try to state about the checklist you used and where you adopt or adapt the checklist or questionnaire. .

5. Result

No need to state repeat similar concepts in different data presentation form. Therefore, it is better to state either in table or graph with brief description, or in text form….

Your logistic regression model is not clear?

o You only conducted bivariable logistic regression which is not enough and didn’t consider the confounders, and your odds ratio is not clear whether it is AOR or COR.

If your odds ratio (OR) is Adjusted Odds Ratio (AOR), where is your COR and what was the criteria to pass your variables from bivariable to multi variable logistic regression?

If your odds ratio (OR) is COR you need to conduct multivariable regression model for better decision since COR is didn’t consider other confounding variables.

If you refine the above things, your study would be valuable. Therefore, try to focus on your logistic regression model.

Reviewer #2: Reviewer comments

Manuscript Number: PONE-D-21-25497

Title "Health hazards related to using masks and, or personal protective equipment among the physicians working in different public hospitals in Dhaka: A cross-sectional study".

Generally speaking:

Thank you for providing me the opportunity to review this manuscript that raises important issues about health hazards related to using masks and/or personal protective equipment in one of the developing countries.

Comment 1:

1. ABSTRACT:

a) Results:

• The number of samples should be mentioned in the methodology of the abstract.

• (35.4[7.7]) these numbers should be clarified.

• Furthermore, the feminine female gender and reusing masks for an extended period were the risk factors for facial disfigurement.

Comment 2:

2. INTRODUCTION:

a) The text of the introduction does not seem to have coherence and integrity. It should be concise and targeted to the aim.

b) Global/ Regional/ Bangladesh prevalence of health hazards related to using of personal protective equipment among health workers should be mentioned. The current situation of other developing and developed countries should also be added.

c) Risk factors associated with health hazards related to using of personal protective equipment should be clearly stated.

d) Explaining why this topic was chosen for analysis in this article is not well written. The benefits of conducting the study to the community should be explained.

Comment 3:

3. METHODS:

Generally, the information mentioned under the methods are too long with redundancy and diffusion and should be divided into several sub-sections as follow:

a) Type of the study

b) Study setting

c) Study Participants: The characteristics of the study participants should be mentioned as inclusion criteria and exclusion criteria (if any)

d) Sample size

e) Study tools:

• There was no clear mention of the questionnaire used. Was it newly developed by the authors (if so, include the reference)? Was it piloted to assess its internal consistencies? Was it validated?

• It is advisable to include the questions as per each domain in the methodology, how to score the questionnaire, specify the cut points in the questionnaire, as well as how long did it take to complete each questionnaire?

f) Data management analysis

g) Ethical considerations

Comment 4:

4. RESULTS:

1) In line 237, I don't agree with the term ''mostly'' when describing those worked in COVID-19 dedicated units, they were about two thirds and not the most of study population. Again, in line 238, the term ''most'' for describing the respondents worked on the roster is not true, they were three quarters and not the most of study population. In line 280, the term ''most'' was repeated twice while the percentages were 52% and 39%. It is advisable to revise the rest of the result comments.

2) How does increasing personal stress level increased the chance of dyspnea with any severity, while OR is 0.7.

3) In line 284, it is table 2 not table 3.

4) Assessment of Relative Risk in table 1 should be explained.

Comment 5:

5. DISCUSSION:

a) Should give reasons for the risk factors. The manuscript could be greatly strengthened if the authors could provide highlight on the risk factors in other developing and developed countries with similar context.

b) Compare the findings of the study with other findings and state the reasons for the strengths and weaknesses in each section.

a) Line 356: “Headache prevalence is similar to other studies”. It is advisable to give the prevalence of headache and other health hazards which was not mentioned, as well as the prevalence of the risk factors.

b) Line 364: regular psychological assessment and counseling for managing the stress

c) The sentence “It was a cross-sectional study” was repeated many times. e.g, it was repeated two times in the last paragraph of the discussion.

Comment 6:

6. CONCLUSION:

It is unclear. It should be specific and based on the findings of the study.

Comment 7:

7. STRENGTHS AND LIMITATIONS:

a) Please analyze the strengths of the study.

b) Last paragraph of discussion is the limitations of the study

Comment 8:

8. REFERENCES:

a) Please revise the Harvard Vancouver method.

b) All references should be written in the same way.

Comment 9:

English needs to be revised.

6. PLOS authors have the option to publish the peer review history of their article (what does this mean?). If published, this will include your full peer review and any attached files.

Reviewer #1: No

Reviewer #2: No

---

## [Author Response · Author response to Decision Letter 0]

23 Apr 2022

Response to Academic editor

1. Why you used the heading “Methods and Materials”? Is there a material that used for this study? if not, please omit the “materials”.

Response: Thank you for the note. There is no material used in this study. It was a mistake. I have omitted materials. (line 150)

2. The methods will be better if structured in this way; Study Area and Period, Study Design, source population, study population, Eligibility criteria, Sample Size Determination, Sampling Technique and Procedures, Study Variables, Operational Definitions, Data Collection Instrument, Data Collection Procedures, Data Quality Control, and Data Processing and Analysis. Specially, this section should be critically addressed!

Response: Thanks for the advice. I have revised the methods as per your instructions.

(Line-154,160,169,172, 178, 182, 212,253,259, 265,277, 288)

3. The authors should clearly explain their Eligibility criteria as; inclusion and exclusion criteria.

Response: mentioned in the specific section (line 172-177)

4. The authors must operationalize their outcome variables. How did they measure their outcome variables?

Response: the definition of the outcome variables was added in the operational definition section and Measurement of dyspnea was made with MRC score and headache with HIT-6 score questionnaire.

(Line: 199-211)

5. Concerning to the sample size calculation for this study, the authors explained as …. So the estimated sample size was 384. We included a total of 506 respondents in this study” why you didn’t used the calculated sample size? This needs a justification.

Response: We were not sure how many of the approached physicians would respond. So, we approached to 1122 physicians and 506 responses were ultimately included in the study. As this line is creating confusion, we deleted this line.

6. How did they get the study subjects? They have to clearly address their sampling technique?

Response: added in the corresponding section (line 182-185)

7. Since you conducted the data collection with online means, how did you keep the data quality for this study?

Response: added in the corresponding section (line-265-276)

8. Study instrument: have you developed, adapted or adopted the questionnaires? You have to clearly explain about the questionnaires you used because this is crucial. You have to report the reliability analysis finding for your questionnaires.

Response: we developed the questionnaire. We made the reliability with Cronbach’s alpha, which was 0.7. and validity test with Pearson correlation. Added in the text. (line-267-279)

9. Add the response rate.

Response: We accessed a total of 1122 doctors; among them, 534 doctors responded. Response rate (47.6%). Added in the text (line 292-293)

10. Discussion: discuss by using the scientific reasoning.

Response: some revision made as per your instruction

11. Conclusion: avoid numbering. Make it a paragraph. Focus on your main findings, clinical implications, recommendations etc.

Response: Corrected in the text as per your instruction

Response to the reviewer:

Thank you for reviewing the article. I have tried to address all the points raised in your review comments.

Reviewer #1: Manuscript Title: Health hazards related to using masks and, or personal protective equipment among the physicians working in different public hospitals in Dhaka: A cross-sectional survey

1. Title: it is researchable, interesting and targeted. The manuscript had presented Health hazards related to using masks and, or personal protective equipment among the physicians working in public hospitals. It can be a valuable scientific literature if published after incorporation of the comments raised.

2. Abstract:

-

3. Introduction

-

4. Methods

Better to state about the brief background of your study area, from how many hospitals the physicians recruited.

Response: Stated (line-157-159)

Why you conduct only on physicians, what about other health care workers like Nurses, pharmacists, midwiferies, Anesthesia…?

Response: We made the case record form in English and we used lot of medical terms. In Bangladesh we thought it would be difficult for the Nurses, pharmacist, midwiferies to understand every aspect of the case record form.

If you conceptual framework it is better to show the relationship of predictor and outcome variables in the form of conceptual framework.

Response: this study is not a conceptual frame work

Better to clarify about your sample size determination, I think you used census method, so what is the advantage of using single population proportion formula?

Response: we included only the physicians. So, we used single population proportion formula

What types of technique you used to keep/control your data quality? Since data quality is the corner stone for mixed research approach.

Response: Thanks for your concern. I have revised the writing and explained it in line-265-276.

I haven’t seen about your tool/questionnaire or checklist you used to collect your data; therefore, please try to state about the checklist you used and where you adopt or adapt the checklist or questionnaire. .

Response: added in the supplement-1

5. Result

No need to state repeat similar concepts in different data presentation form. Therefore, it is better to state either in table or graph with brief description, or in text form….

Response: Thank you for your advice. I have corrected as much as possible.

Your logistic regression model is not clear?

o You only conducted bivariable logistic regression which is not enough and didn’t consider the confounders, and your odds ratio is not clear whether it is AOR or COR.

If your odds ratio (OR) is Adjusted Odds Ratio (AOR), where is your COR and what was the criteria to pass your variables from bivariable to multi variable logistic regression?

If your odds ratio (OR) is COR you need to conduct multivariable regression model for better decision since COR is didn’t consider other confounding variables.

Response: It was adjusted odd ratio. We used forward conditional methods in the regression model. Crude odd ration was added in the supplement 3. (line 342-344)

If you refine the above things, your study would be valuable. Therefore, try to focus on your logistic regression model.

Reviewer #2: Reviewer comments

Manuscript Number: PONE-D-21-25497

Title "Health hazards related to using masks and, or personal protective equipment among the physicians working in different public hospitals in Dhaka: A cross-sectional study".

Generally speaking:

Thank you for providing me the opportunity to review this manuscript that raises important issues about health hazards related to using masks and/or personal protective equipment in one of the developing countries.

Comment 1:

1. ABSTRACT:

a) Results:

• The number of samples should be mentioned in the methodology of the abstract.

Response: Added (line-80)

• (35.4[7.7]) these numbers should be clarified.

Response: It was mean (SD) age, corrected (Line-83)

• Furthermore, the feminine female gender and reusing masks for an extended period were the risk factors for facial disfigurement.

Response: Rewritten as “Furthermore, the female gender, reusing masks for an extended period (> 6 hours) were the risk factors for facial disfigurement.”

Comment 2:

2. INTRODUCTION:

a) The text of the introduction does not seem to have coherence and integrity. It should be concise and targeted to the aim.

b) Global/ Regional/ Bangladesh prevalence of health hazards related to using of personal protective equipment among health workers should be mentioned. The current situation of other developing and developed countries should also be added.

Response: Thank you for your advice. I have rewritten this section, tried to be concise and targeted as per your advice. But we found very few comprehensive studies in this regard. So current situation in the developed and developing countries cannot be added. I believe it would be better if we could add it.

c) Risk factors associated with health hazards related to using of personal protective equipment should be clearly stated.

Response: Added in the introduction to some extent, according to other studies. (line-137-139)

d) Explaining why this topic was chosen for analysis in this article is not well written. The benefits of conducting the study to the community should be explained.

Response: Rewritten. Line-146-149

Comment 3:

3. METHODS:

Generally, the information mentioned under the methods are too long with redundancy and diffusion and should be divided into several sub-sections as follow:

a) Type of the study

b) Study setting

c) Study Participants: The characteristics of the study participants should be mentioned as inclusion criteria and exclusion criteria (if any)

d) Sample size

e) Study tools:

• There was no clear mention of the questionnaire used. Was it newly developed by the authors (if so, include the reference)? Was it piloted to assess its internal consistencies? Was it validated?

• It is advisable to include the questions as per each domain in the methodology, how to score the questionnaire, specify the cut points in the questionnaire, as well as how long did it take to complete each questionnaire?

f) Data management analysis

g) Ethical considerations

Response: rewritten as per instruction (line-154,160,169,172, 178, 182, 212,253,259, 265,277, 288)

Comment 4:

4. RESULTS:

1) In line 237, I don't agree with the term ''mostly'' when describing those worked in COVID-19 dedicated units, they were about two thirds and not the most of study population. Again, in line 238, the term ''most'' for describing the respondents worked on the roster is not true, they were three quarters and not the most of study population. In line 280, the term ''most'' was repeated twice while the percentages were 52% and 39%. It is advisable to revise the rest of the result comments.

Response: Thank you for your advice I have corrected these. (line- 299, 300, 301 etc)

2) How does increasing personal stress level increased the chance of dyspnea with any severity, while OR is 0.7.

Response: corrected (Line-348-349)

3) In line 284, it is table 2 not table 3.

Response: Corrected

4) Assessment of Relative Risk in table 1 should be explained.

Response: I have rewritten it and explained in 309-323

Comment 5:

5. DISCUSSION:

a) Should give reasons for the risk factors. The manuscript could be greatly strengthened if the authors could provide highlight on the risk factors in other developing and developed countries with similar context.

Response: Thank you for your advice. I have rewrite tis section, tried to focused on your concern Limited literature was found even after extensive search to compare. 

b) Compare the findings of the study with other findings and state the reasons for the strengths and weaknesses in each section.

Response: Tried to address as far as possible. We found a few study to compare. 

a) Line 356: “Headache prevalence is similar to other studies”. It is advisable to give the prevalence of headache and other health hazards which was not mentioned, as well as the prevalence of the risk factors.

Response: Corrected (line 419, 440, 

b) Line 364: regular psychological assessment and counseling for managing the stress

Response: Corrected

c) The sentence “It was a cross-sectional study” was repeated many times. e.g, it was repeated two times in the last paragraph of the discussion.

Response: Corrected

Comment 6:

6. CONCLUSION:

It is unclear. It should be specific and based on the findings of the study.

Response: The writing of the conclusion was revised.

Comment 7:

7. STRENGTHS AND LIMITATIONS:

a) Please analyze the strengths of the study.

b) Last paragraph of discussion is the limitations of the study

Response: Added in the text

Comment 8:

8. REFERENCES:

a) Please revise the Harvard Vancouver method.

b) All references should be written in the same way.

Response: I have revised as per your instruction

Comment 9:

English needs to be revised.

Response: The manuscript has been revised by Editage for English language

---

## [Decision Letter · Decision Letter 1]

14 Jul 2022

PONE-D-21-25497R1Health hazards related to using masks and/or personal protective equipment among physicians working in public hospitals in Dhaka: a cross-sectional studyPLOS ONE

Dear Dr. Mahmud,

Thank you for submitting your manuscript to PLOS ONE. After careful consideration, we feel that it has merit but does not fully meet PLOS ONE’s publication criteria as it currently stands. Therefore, we invite you to submit a revised version of the manuscript that addresses the points raised during the review process. While Reviewer #2 sees all points addressed, Reviewer #3 still has some minor comments. Please submit your revised manuscript by Aug 28 2022 11:59PM. If you will need more time than this to complete your revisions, please reply to this message or contact the journal office at plosone@plos.org. Please include the following items when submitting your revised manuscript:

We look forward to receiving your revised manuscript.

Kind regards,

Christoph Strumann

Academic Editor

PLOS ONE

Journal Requirements:

Reviewers' comments:

Reviewer's Responses to Questions

**Comments to the Author**

1. If the authors have adequately addressed your comments raised in a previous round of review and you feel that this manuscript is now acceptable for publication, you may indicate that here to bypass the “Comments to the Author” section, enter your conflict of interest statement in the “Confidential to Editor” section, and submit your "Accept" recommendation.

Reviewer #2: All comments have been addressed

Reviewer #3: (No Response)

2. Is the manuscript technically sound, and do the data support the conclusions?

Reviewer #2: Yes

Reviewer #3: Yes

3. Has the statistical analysis been performed appropriately and rigorously? 

Reviewer #2: Yes

Reviewer #3: No

4. Have the authors made all data underlying the findings in their manuscript fully available?

Reviewer #2: Yes

Reviewer #3: Yes

5. Is the manuscript presented in an intelligible fashion and written in standard English?

Reviewer #2: Yes

Reviewer #3: Yes

6. Review Comments to the Author

Reviewer #2: Reviewer comments

Manuscript Number: PONE-D-22-03316R1

Title "Health hazards related to using masks and, or personal protective equipment among the physicians working in different public hospitals in Dhaka: A cross-sectional study".

Thank you for providing me the opportunity to review this manuscript that raises important issues about health hazards related to using masks and/or personal protective equipment in one of the developing countries.

It seems that all corrections were done.

Reviewer #3: The authors of this manuscript have improved upon its quality substantially based on the comments from the first review. However, the authors should address the comments raised below:

Line 121: Change these words "As frontliners of this pandemic," to read "As frontliners in the control of this pandemic,"

Line 124: Are there updated/more recent figures on the mortality of HCWs due to COVID-19? Since this study focused on physicians only, are there specific records for physicians only?

Line 154: What are the reasons for selecting these public hospitals? How many public hospitals are in Bangladesh? Are these hospitals representative enough? A map of Bangladesh showing the sampled hospitals would be a good fit.

Lines 172-177: As part of the eligibility criteria, the authors should indicate within the manuscript the reason why other HCWs were excluded from the study.

Lines 178-181: Authors should include a reference for using the stated formula in calculating the sample size. Furthermore, the authors must include a justification within the manuscript as to the reason why respondents well-above the calculated sample size were recruited into the study.

Line 182: I believe the authors have used a non-probability sampling technique "Purposive" since physicians were targeted. It won't be out of place to indicate that a Purposive sampling technique was used.

Line 284: Why have the authors considered the use of Relative risk and not odds ratio in this cross-sectional study?

Table 1: Authors should include among the keys below the table the full terms of all abbreviations used.

Table 3: Authors should include among the keys below the table the full terms of all abbreviations used.

The authors should include a paragraph just before the conclusion stating clearly the limitations associated with this study and how these limitations affect the generalizability of their findings.

7. PLOS authors have the option to publish the peer review history of their article (what does this mean?). If published, this will include your full peer review and any attached files.

Reviewer #2: No

Reviewer #3: **Yes: **Ismail Ayoade Odetokun (Ph.D.)

---

## [Author Response · Author response to Decision Letter 1]

23 Aug 2022

In response to review of the manuscript entitled “Health hazards related to using masks and/or personal protective equipment among physicians working in public hospitals in Dhaka: a cross-sectional study”.

Dear Sir,

Thank you for reviewing my manuscript. I have tried to address each point raised by the academic editor and the reviewers.

Response to Academic editor:

Thank you for giving me the opportunity to revise the article.

Response to the reviewer:

Thank you for reviewing the article. I have tried to address all the points raised in your review comments.

Reviewer #2: 

Thank you for providing me the opportunity to review this manuscript that raises important issues about health hazards related to using masks and/or personal protective equipment in one of the developing countries.

It seems that all corrections were done.

Response: Thank you for accepting the corrections

Reviewer #3: 

The authors of this manuscript have improved upon its quality substantially based on the comments from the first review. However, the authors should address the comments raised below:

Line 121: Change these words "As frontliners of this pandemic," to read "As frontliners in the control of this pandemic,"

Response: Thank you for the correction. I have corrected as per your advice.

Line 124: Are there updated/more recent figures on the mortality of HCWs due to COVID-19? Since this study focused on physicians only, are there specific records for physicians only?

Response: 

I have searched extensively for updated/recent figures, but failed to find updated figures on physicians only. 

I have found an estimate of WHO, So I added the following statement in the manuscript “Reported death to WHO COVID-19 surveillance in health care workers due to COVID -19, from January 2020–May 2021 was only 6643. Still, a population-based estimate revealed that 115 493 health and care workers could have died from COVID-19 during that period”

.Line 154: What are the reasons for selecting these public hospitals? How many public hospitals are in Bangladesh? Are these hospitals representative enough? A map of Bangladesh showing the sampled hospitals would be a good fit.

Response: There are 3,976 healthcare facilities in the public sector in Bangladesh; It includes union subcenters. Among them, 600 facilities were dedicated to treating COVID-19 patients nationwide. We did the study in Dhaka city, the capital of Bangladesh. In Dhaka city, nine large public hospitals were involved in COVID treatment. Among them, we choose these five hospitals purposively. These hospitals are representative enough of Dhaka city. We did not include other health facilities nationwide, as health facilities outside Dhaka are not homogenous, and there is a communication problem.

To avoid redundancy, we did not mention all this information in the manuscript. We corrected the statement below” Among nine large COVID-19 dedicated hospitals in Dhaka city, we purposively selected these five hospitals.” A Map of Dhaka showing this hospital has been attached as supplements. 

Lines 172-177: As part of the eligibility criteria, the authors should indicate within the manuscript the reason why other HCWs were excluded from the study.

Response: added in 195-197.

Lines 178-181: Authors should include a reference for using the stated formula in calculating the sample size. Furthermore, the authors must include a justification within the manuscript as to the reason why respondents well-above the calculated sample size were recruited into the study.

Response: Reference added line-199, Justification of recruiting repondents well above the calculated sample size added, line-201-203. 

Line 182: I believe the authors have used a non-probability sampling technique "Purposive" since physicians were targeted. It won't be out of place to indicate that a Purposive sampling technique was used.

Response: we deleted the line

Line 284: Why have the authors considered the use of Relative risk and not odds ratio in this cross-sectional study?

Response: Thanks for pointing this. It was a mistake, so we corrected it. Line 318.

Table 1: Authors should include among the keys below the table the full terms of all abbreviations used.

Table 3: Authors should include among the keys below the table the full terms of all abbreviations used.

Response: Added in the manuscript

The authors should include a paragraph just before the conclusion stating clearly the limitations associated with this study and how these limitations affect the generalizability of their findings

Response: added in 604-621 lines. 

Journal Requirements:

I have reviewed the references and found one reference might be retracted one, so deleted that reference, number 32. 

“Warren DW, Mayo R, Zajac DJ, Rochet AH. Dyspnea following experimentally induced increased nasal airway resistance. Cleft Palate Craniofac J. 1996 May;33(3):231-5. doi: 10.1597/1545-1569_1996_033_0231_dfeiin_2.3.co_2. PMID: 8734724.”

I have added two more references as per requirement of the reviewers

No-8: World Health Organization‎. The impact of COVID-19 on health and care workers: a closer look at deaths. World Health Organization. 2021, Accessed on 10.08.2022. https://apps.who.int/iris/handle/10665/345300.

No-16: Daniel WW, editor. 7th ed. New York: John Wiley & Sons; 1999. Biostatistics: a foundation for analysis in the health sciences.

I hope I have tried my level best to address all of your point raised during the review process. Please consider my manuscript for publication in PLOS ONE.

Thanks

Dr. Reaz Mahmud

Assistant professor Neurology

Dhaka Medical College

---

## [Editor Report · Decision Letter 2]

24 Aug 2022

Health hazards related to using masks and/or personal protective equipment among physicians working in public hospitals in Dhaka: a cross-sectional study

PONE-D-21-25497R2

Dear Dr. Mahmud,

We’re pleased to inform you that your manuscript has been judged scientifically suitable for publication and will be formally accepted for publication once it meets all outstanding technical requirements.

Kind regards,

Christoph Strumann

Academic Editor

PLOS ONE
---

## [Editor Report · Acceptance letter]

2 Sep 2022

PONE-D-21-25497R2 

Health hazards related to using masks and/or personal protective equipment among physicians working in public hospitals in Dhaka: a cross-sectional study 

Dear Dr. Mahmud:

I'm pleased to inform you that your manuscript has been deemed suitable for publication in PLOS ONE. Congratulations! Your manuscript is now with our production department. 

Kind regards, 

on behalf of

Dr. Christoph Strumann 

Academic Editor

PLOS ONE